# Effects of exercise modalities on heart rate recovery and its association with cardiometabolic risk in adolescents with overweight or obesity

Rubin Pooni[1], Silva Arslanian[2], Heather Edgell[1], Hala Tamim[1], SoJung Lee[3,4]*, Jennifer L. Kuk[1]*

**1** School of Kinesiology and Health Science, York University, Toronto, Ontario, Canada, **2** Center for Pediatric Research in Obesity and Metabolism and the Division of Pediatric Endocrinology, Metabolism, and Diabetes Mellitus, UPMC Children's Hospital of Pittsburgh, University of Pittsburgh School of Medicine, Pittsburgh, Pennsylvania, United States of America, **3** Division of Sports Medicine and Science, Graduate School of Physical Education, Kyung Hee University, Yongin, Republic of Korea, **4** Obesity and Physical Activity Research Laboratory, Kyung Hee University, Yongin, Republic of Korea

* jennkuk@yorku.ca (JLK); sojung.lee@khu.ac.kr (SL)

## Abstract

### Objective

Attenuated heart rate recovery (HRR) immediately after exercise is an independent predictor of cardiovascular disease and mortality in adults. We examined the effects of aerobic exercise (AE), resistance exercise (RE), and combined AE and RE on HRR, and the relationship of HRR with body composition and metabolic risk factors in adolescents with overweight or obesity.

### Research design and methods

We included 147 adolescents (BMI ≥ 85th percentile, 12–18 years) who participated in exercise intervention studies (3–6 months), and had a complete data set including $VO_{2peak}$, body composition by dual-energy X-ray absorptiometry, and cardiometabolic risk factors before and after the interventions. HRR was calculated as the difference between peak HR during the maximal treadmill test and HR at 1-, 2-, 3-, 4-, and 5-min after the cessation of the test.

### Results

After interventions, a faster HRR at 2–5 min was observed following AE (17.3–25.6% change, $P < 0.0001$), RE (7.1–10.9%, $P < 0.05$), and combined AE and RE (10.7–12.0%, $P < 0.05$) compared to pre-intervention. Compared to controls, AE (14.0–24.0%, $P < 0.05$) was the only exercise group to exhibit a significantly faster HRR at 1-, 2-, 3-, 4-, and 5-min after the exercise intervention. Collapsed across exercise groups, improvements in 2-, 3-, 4-, and 5-min HRR are independently associated

**Data availability statement:** Under the terms of the original informed consent, posting participant data in a public depository would be a breach of the approved ethics protocol approved by the University of Pittsburgh Institutional Review Board. Data can be shared with ethical approval from the requestors relevant ethical review board. Data requests can be sent to Dr. SoJung Lee (sojung.lee@khu.ac.kr) or the Associate Dean, Research and Innovation at York University (Dr. Chris Ardern - hhadri@yorku.ca).

**Funding:** The current analyses received no specific funding. The original research was funded by the National Institutes of Health (5R01HL114857, 1R21DK083654-01A1), American Diabetes Association (7-08-JF-27), UPMC Children's Hospital of Pittsburgh (Cochrane-Weber Foundation and Renziehausen Fund) to Lee, and the National Center for Advancing Translational Sciences Clinical and Translational Science Award (UL1 RR024153, UL1TR000005) to the Pediatric Clinical and Translational Research Center at UPMC Children's Hospital of Pittsburgh. The funders had no role in study design, data collection and analysis, decision to publish, or preparation of the manuscript.

**Competing interests:** The authors have declared that no competing interests exist.

($P < 0.05$) with increases in $VO_{2peak}$. Changes in HRR were not associated with the changes in % body fat or metabolic risk factors.

## Conclusion

AE training is more beneficial than RE or combined AE and RE training for improving HRR in adolescents with overweight or obesity.

## Introduction

Heart rate (HR) is determined by the intrinsic activity of the sinus node, which is modulated by the autonomic nervous system [1]. The rise in heart rate (HR) during exercise is mainly due to decreased parasympathetic activity and increased sympathetic activity [1]. As exercise intensity increases, sympathetic activity plays a greater role in increasing HR [2]. In contrast, the reduction in HR immediately after exercise is attributed to parasympathetic reactivity and sympathetic withdrawal [3].

Heart rate recovery (HRR) is the rate at which heart rate declines following the cessation of exercise. Previous studies have shown that a delayed HRR is a marker of impaired parasympathetic reactivity [4], and an independent predictor of all-cause mortality, sudden cardiac death, and a risk factor for cardiovascular disease and type 2 diabetes in adults [5–9]. Impaired HRR after exercise has been associated with higher body mass index (BMI) and indices of obesity in adults [10–12]. Similarly, attenuated HRR has been observed in adolescents with overweight and obesity [13–15].

While impaired HRR is well documented in individuals with a higher BMI, few studies have examined the effects of regular exercise on HRR in this population. Those studies that do exist have demonstrated that moderate-intensity aerobic training (AE) is associated with improved HRR in adults with overweight or obesity, metabolic syndrome and obstructive sleep apnea, though not all studies demonstrate changes relative to a control group [16–19]. Further, Kim et al. [17] observed improvements in HRR only in men with metabolic syndrome and not in those without. Currently, it is unknown whether other types of exercise, such as resistance training (RE) or a combination of AE and RE, would also be effective for improving HRR, or if these observations hold true in adolescent populations. Therefore, we examined the effects of AE, RE, and combined AE and RE on HRR, and examined the relationship of HRR with body composition and cardiometabolic risk factors in adolescents with overweight and obesity.

## Materials and methods

### Participants

This is a secondary data analysis of our previously published randomized trials of AE, RE, and combined AE and RE for 3–6 months (Clinicaltrials.gov registration numbers: NCT00739180, NCT01323088, NCT01938950) conducted between 2007 and 2017 at UPMC Children's Hospital of Pittsburgh [20–22]. The original primary aims

of these studies [20–22] were to examine the effects of different exercise training on body composition and insulin resistance in adolescents. Among 207 adolescents with overweight and obesity (12–18 years, BMI ≥ 85th percentile) randomized for these trials [20–22], participants were excluded from the current analyses if they were missing all 5 min of HRR at pre-intervention ($n = 4$), all 5 min of HRR at post-intervention ($n = 47$), or covariate data ($n = 9$), leaving a final sample of 147 participants for the current investigation (Fig 1). The inclusion criteria were that participants be 12–18 years of age with overweight or obesity (BMI ≥ 85th percentile) [23], pubertal (Tanner stages II-V; i.e., genital development and pubic hair [24]), inactive (i.e., no structured physical activity for 3 months prior to the study except school physical education classes), and non-smokers. The exclusion criteria were significant weight change (BMI > 2–3 kg/m$^2$) within 3 months prior to the study, having an endocrine disorder, or taking medication influencing body composition, lipids, or glucose metabolism [20–22]. All participants provided written parental informed consent and child assent prior to participation in the study. A certified nurse practitioner conducted a physical examination and identified pubertal development according to Tanner criteria. Participants self-identified their ethnicity and were categorized as being of white ethnicity or non-white ethnicity for analysis. The investigation was approved by the University of Pittsburgh Institutional Review Board. To reduce confirmation bias and opposing evidence, there were no specific hypotheses generated for this investigation.

## Anthropometrics

Body weight was measured and recorded to the nearest 0.1 kg using a standing scale (Befour Medichoice model SCAL31MC, Saukville, WI, USA) and height was measured to the nearest 0.1 cm using a Harpenden stadiometer

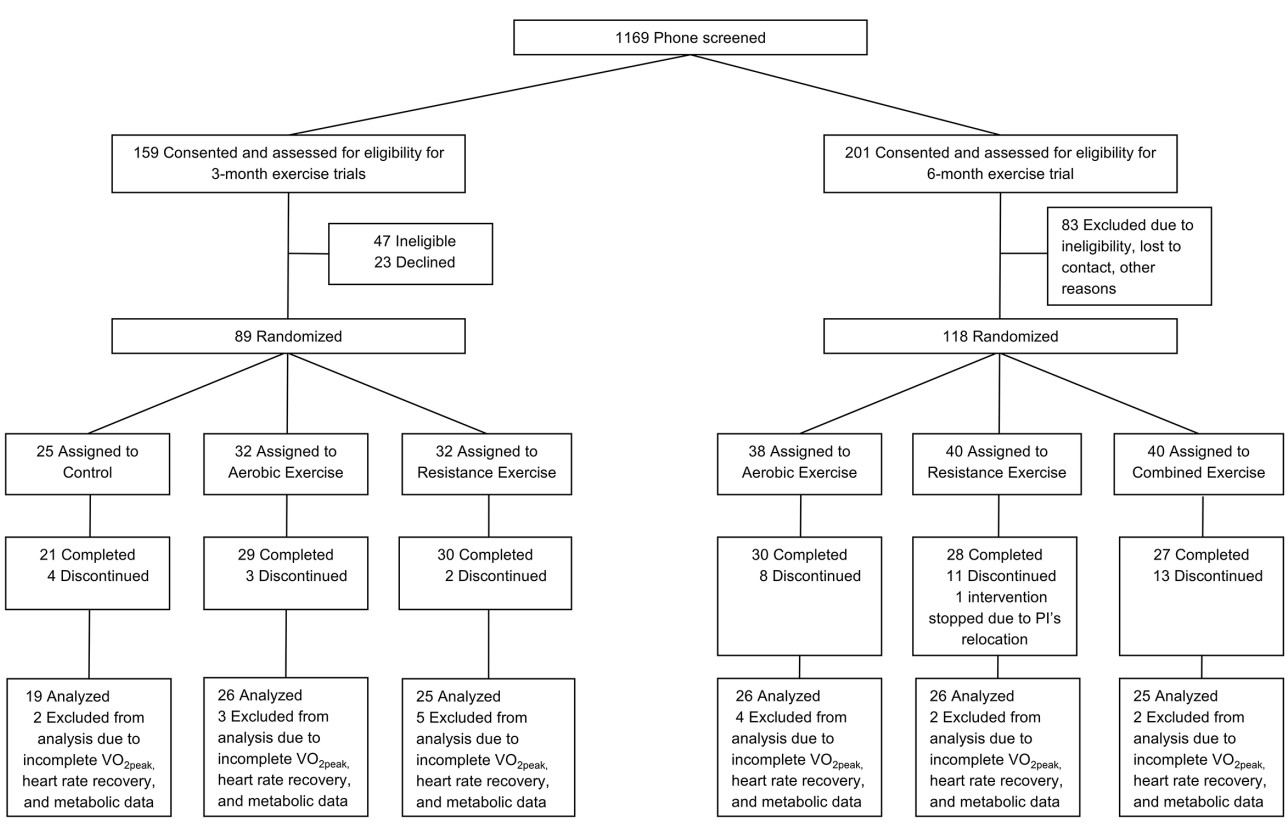

**Fig 1. Participant flow diagram.**

(Holtain Ltd, Wales, UK). Height and weight measurements were used to calculate BMI using the following equation: BMI ($kg/m^2$) = weight (kg)/ height ($m^2$). Waist circumference was measured in a standing position at the superior edge of the iliac crest after a normal expiration to the nearest 0.1 cm and the average of two values was used for analysis. Body composition was assessed by dual-energy X-ray absorptiometry using Lunar iDXA (GE Healthcare, Madison, Wisconsin, USA) [25].

## Cardiometabolic risk factors

A fasting blood sample was obtained for lipid analysis following a minimum 10-hour overnight fast. Fasting plasma lipids were measured using an Olympus AU400 Chemistry Analyzer in the Nutrition Laboratory of the University of Pittsburgh Graduate School of Public Health, which was certified by the Centers for Disease Control and Prevention-National Heart, Lung, and Blood Institute Lipid Standardization Program [26]. HDL was assessed using the two-reagent method. Triglycerides were determined enzymatically. Reagents used were obtained from Vital Diagnostics (Lincoln, RI). LDL and VLDL were calculated using the Friedewald equation [27]. For total and HDL cholesterol and triglycerides intra-assay coefficients of variation (CV) were 1.0%, 1.8% and 1.8% and inter-assay CV were 1.6%, 2.6%, and 3.7%, respectively.

## Assessment of $VO_{2peak}$ and HRR

Cardiorespiratory fitness was assessed using a modified Balke maximal treadmill test with the use of standard open-circuit spirometry techniques until volitional fatigue [20–22]. The treadmill was set at a brisk but comfortable walking speed and was held constant for the duration of the test. For the first 2 minutes of the test, the treadmill grade was set at 0%, then increased to 2% for the third minute, with a 1% increase every minute thereafter until exhaustion. The MOXUS (AEI Technologies, Pittsburgh, PA, USA) gas analyzer was used in the 3-month studies [21,22], and the True One 2400 (Parvo Medics, Sandy, UT, USA) gas analyzer was used in the 6-month intervention study [20] to measure oxygen consumption. $VO_{2peak}$ was defined as the average peak oxygen consumption in the last 20 seconds of the maximal treadmill test.

After completing the maximal treadmill test, HR data was recorded for 5 minutes while participants performed a 5-min cool-down on the treadmill at 2 mph with a 0% grade. HR was recorded continuously during the test and the 5-min cool-down using a HR monitor (Polar Oy, Kempele, Finland). The HRR was calculated as the difference between the peak HR at near-maximal treadmill stages and the HR taken at 1-, 2-, 3-, 4-, and 5-min after the completion of the test.

## Exercise interventions

Participants were randomly assigned to one of the following groups: AE, RE, combined AE and RE, or a non-exercise control group as reported previously [20–22]. Participants in the AE group exercised on a treadmill and/or elliptical at moderate-intensity (50%−75% of $VO_{2peak}$), 3 sessions/week (60 minutes/session), for 3–6 months [20–22]. Participants in the RE group completed push-ups, abdominal crunches and 8 resistance exercises (leg press, leg extension, leg flexion, chest press, latissimus pull-down, seated row, bicep curl, triceps extension) on weight machines, 3 sessions per week (60 minutes per session). The participants performed 2 sets of 8–12 repetitions per set during the 3-month study [21–22], and 2 sets of 12–15 repetitions per set during the 6-month study [20]. Participants in the combined AE and RE group [20] completed 30 minutes of moderate-intensity aerobic exercise on a treadmill and/or elliptical and did a series of 8 whole-body resistance exercises (1 set, 12–15 repetitions), 3 sessions/week (60 minutes/session), along with push-ups and abdominal crunches. The control group in the 3 month studies [21–22] were asked not to participate in structured physical activity. Average attendance at the exercise sessions was ≥ 90% in all studies [20–22]. During the intervention period, participants in all three trials [20–22], including controls, were asked to follow a healthy weight maintenance diet (55–60% carbohydrate, 15–20% protein, and 25–30% fat).

## Statistical analyses

Baseline characteristics were stratified by exercise training group. One-way analysis of variance (ANOVA) with Tukey's post-hoc tests and chi-square tests were used to determine between-group differences for continuous and categorical characteristics, respectively. A repeated measures analysis of covariance (ANCOVA) adjusting for sex, Tanner stage, ethnicity, and training duration was used to examine differences in HRR compared to control, and pre-post intervention differences in HRR using proc mixed. Multiple linear regression models were used to examine the relationship between the change in HRR and the changes in $VO_{2peak}$, body weight, waist circumference, % body fat, and cardiometabolic risk factors in all exercise groups collapsed together. All regression models were adjusted for sex, Tanner stage, ethnicity, training duration, and types of training. There was no main effect or interaction effect for training duration, so data from 3- and 6-month training interventions were combined together for analysis. Deterministic regression imputation was used to calculate missing HRR values for analysis. All statistical analyses were performed using SAS statistical software (version 9.4; SAS Institute, Cary, NC) with significance defined at $P < 0.05$.

## Results

### Participant characteristics

Participant characteristics at baseline stratified by exercise group are presented in Table 1. There were no significant differences in age, sex, Tanner stage, ethnicity, body weight, waist circumference, % body fat, $VO_{2peak}$, total cholesterol, HDL cholesterol, or LDL cholesterol between groups ($P > 0.05$). The combined AE and RE group had a significantly lower

**Table 1. Participant Characteristics.**

| Variable | Control (n=19) | Aerobic Exercise (n=52) | Resistance Exercise (n=51) | Aerobic + Resistance (n=25) | P value |
|---|---|---|---|---|---|
| **Demographic Characteristics** | | | | | |
| Age (years) | 14.7±1.7 | 14.7±1.8 | 14.5±1.6 | 14.7±1.7 | 0.90 |
| Male (n, %) | 11 (58) | 20 (38) | 26 (51) | 9 (36) | 0.29 |
| Tanner Stage, II-IV/ V (n) | 4/15 | 18/34 | 19/32 | 7/18 | 0.57 |
| White Ethnicity (n, %) | 8 (42) | 27 (52) | 21 (41) | 8 (32) | 0.40 |
| **Aerobic Fitness** | | | | | |
| VO₂peak (mL/kg/min) | 28.3±4.7 | 26.3±4.6 | 25.6±5.4 | 24.7±4.5 | 0.09 |
| **Body Composition** | | | | | |
| Body Weight (kg) | 97.9±14.2 | 95.1±18.6 | 93.4±14.0 | 88.7±15.3 | 0.25 |
| BMI (kg/m²) | 34.7±4.5 | 33.9±4.8 | 34.0±3.2 | 31.4±3.9ᵃ | 0.03 |
| BMI (percentile) | 98.7±1.2 | 98.2±1.7 | 98.6±1.0 | 97.0±3.0ᵃ,ᵇ,ᶜ | 0.001 |
| Waist Circumference (cm) | 110.7±11.6 | 108.9±11.8 | 109.2±9.2 | 104.5±11.2 | 0.22 |
| Body Fat (%) | 42.8±7.0 | 41.7±4.9 | 42.6±4.6 | 40.7±5.6 | 0.41 |
| **Lipids** | | | | | |
| Total Cholesterol (mg/dL) | 146.6±21.4 | 155.1±32.0 | 142.3±24.4 | 143.4±22.3 | 0.08 |
| Triglycerides (mg/dL) | 85.3±36.4 | 115.8±84.1 | 95.0±44.3 | 66.1±30.6ᵇ | 0.007 |
| HDL Cholesterol (mg/dL) | 40.6±6.6 | 43.7±10.6 | 41.1±7.3 | 45.2±9.2 | 0.16 |
| LDL Cholesterol (mg/dL) | 88.9±20.5 | 88.8±25.3 | 82.2±21.5 | 84.9±24.1 | 0.49 |
| VLDL Cholesterol (mg/dL) | 17.1±7.3 | 23.2±16.8 | 19.0±8.9 | 13.2±6.1ᵇ | 0.007 |

Data are presented as mean±SD or frequency (%). P values are displayed for comparison between intervention groups. ᵃ $P < 0.05$ compared to Control. ᵇ $P < 0.05$ compared to Aerobic. ᶜ $P < 0.05$ compared to Resistance.

VO₂peak, peak oxygen uptake; BMI, body mass index; HDL, high-density lipoprotein; LDL, low-density lipoprotein; VLDL, very low-density lipoprotein.

baseline BMI than the control group (*P*=0.03), and a significantly lower baseline BMI percentile than the AE, RE, and control groups (*P*=0.001). The combined AE and RE group had significantly lower baseline triglycerides and VLDL cholesterol than the AE group (*P*=0.007).

### Changes in HRR after exercise intervention

After the exercise interventions, the AE (17.3–25.6% change, *P*<0.0001), RE (7.1–10.9%, *P*<0.05), and combined AE and RE (10.7–12.0%, *P*<0.05) groups all showed a significantly faster 2-, 3-, 4-, and 5-min HRR (Fig 2, Table 2) following the maximal treadmill test, while no significant changes in HRR were observed in the control group. After intervention, the AE (36.4%, *P*<0.0001) and RE (12.0%, *P*=0.02) groups also had a significantly faster HRR at 1-min compared to pre-intervention (Fig 2, Table 2). However, as compared to the control group, the AE group was the only exercise group to exhibit a significantly faster 1- (24.0%, *P*=0.02), 2- (14.0%, *P*=0.02), 3- (16.0%, *P*=0.004), 4- (15.1%, *P*=0.002), and 5-min (16.1%, *P*=0.001) HRR after intervention (Fig 2, Table 2).

### Associations between changes in HRR and VO$_{2peak}$ after exercise intervention

There was no association between the change in 1-min HRR and the change in VO$_{2peak}$ in response to exercise training with or without adjustment for covariates (e.g., sex, Tanner stage, ethnicity, and training duration) and intervention group (Table 3, *P*>0.05). Collapsed across exercise groups, improvements in 2-, 3-, 4-, and 5-min HRR are independently associated with increases in VO$_{2peak}$ after accounting for covariates and the exercise intervention group (*P*<0.0001).

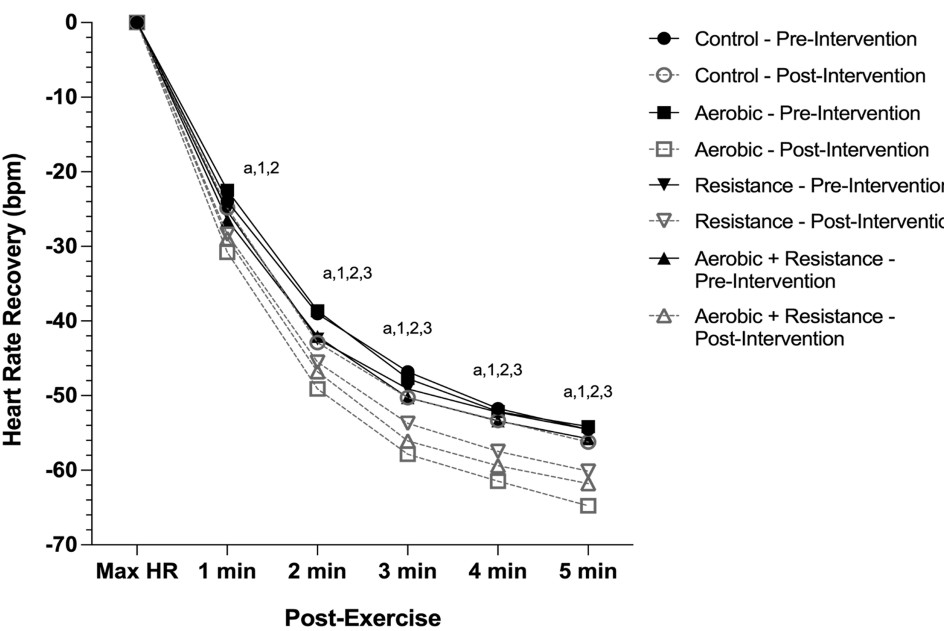

**Fig 2. Heart rate recovery for adolescents with overweight or obesity before and after 3 or 6 months of aerobic exercise (*n*=52), resistance exercise (*n*=51), combined aerobic and resistance exercise (*n*=25), or non-exercise control (*n*=19).** Model was adjusted for sex, Tanner stage, ethnicity, and training duration. Data are presented as adjusted least squared means. [a] *P*<0.05 Aerobic post-intervention versus Control post-intervention; [1] *P*<0.05 Aerobic Pre-intervention versus post-intervention; [2] *P*<0.05 Resistance Pre-intervention versus post-intervention; [3] *P*<0.05 Aerobic+Resistance Pre-intervention versus post-intervention; There were no pre-post intervention differences in the control group.

**Table 2. Heart rate recovery for adolescents with overweight or obesity before and after 3 or 6 months of aerobic exercise (*n* = 52), resistance exercise (*n* = 51), combined aerobic and resistance exercise (*n* = 25), or non-exercise control (*n* = 19). Difference and percentage change in heart rate recovery are presented for pre-intervention versus post-intervention and post-intervention versus control post-intervention.**

| | HRR Pre-Intervention (bpm) | HRR Post-Intervention (bpm) | Pre-Intervention versus Post-Intervention | | | Post-Intervention versus Control Post-Intervention | | |
|---|---|---|---|---|---|---|---|---|
| | | | Difference in HRR (bpm) | Percentage Change in HRR (%) | *P* value | Difference in HRR (bpm) | Percentage Change in HRR (%) | *P* value |
| **Control** | | | | | | | | |
| 1-min HRR | −24 (−20, −28) | −25 (−20, −29) | 1 (−3, 5) | 4.2 | 0.69 | -- | -- | -- |
| 2-min HRR | −39 (−35, −43) | −43 (−38, −47) | 4 (−1, 8) | 10.3 | 0.08 | -- | -- | -- |
| 3-min HRR | −47 (−42, −51) | −50 (−46, −55) | 3 (−1, 8) | 6.4 | 0.12 | -- | -- | -- |
| 4-min HRR | −52 (−47, −56) | −53 (−49, −58) | 2 (−3, 6) | 3.8 | 0.47 | -- | -- | -- |
| 5-min HRR | −55 (−50, −59) | −56 (−52, −61) | 2 (−3, 6) | 3.6 | 0.45 | -- | -- | -- |
| **Aerobic** | | | | | | | | |
| 1-min HRR | −22 (−20, −25) | −31 (−28, −33) | 8 (6, 11) | 36.4 | <0.0001 | 6 (1, 11) | 24.0 | 0.02 |
| 2-min HRR | −39 (−36, −41) | −49 (−47, −52) | 10 (8, 13) | 25.6 | <0.0001 | 6 (1, 11) | 14.0 | 0.02 |
| 3-min HRR | −48 (−45, −50) | −58 (−55, −60) | 10 (8, 13) | 20.8 | <0.0001 | 8 (2, 13) | 16.0 | 0.004 |
| 4-min HRR | −52 (−50, −55) | −61 (−59, −64) | 9 (7, 12) | 17.3 | <0.0001 | 8 (3, 13) | 15.1 | 0.002 |
| 5-min HRR | −54 (−52, −57) | −65 (−62, −67) | 11 (8, 13) | 20.4 | <0.0001 | 9 (3, 14) | 16.1 | 0.001 |
| **Resistance** | | | | | | | | |
| 1-min HRR | −25 (−23, −28) | −28 (−26, −31) | 3 (0.4, 6) | 12.0 | 0.02 | 4 (−2, 9) | 16.0 | 0.18 |
| 2-min HRR | −42 (−40, −45) | −46 (−43, −48) | 3 (0.4, 6) | 7.1 | 0.02 | 3 (−3, 8) | 7.0 | 0.32 |
| 3-min HRR | −49 (−47, −52) | −54 (−51, −56) | 5 (2, 7) | 10.2 | 0.0007 | 3 (−2, 9) | 6.0 | 0.19 |
| 4-min HRR | −52 (−50, −55) | −57 (−55, −60) | 5 (3, 8) | 9.6 | 0.0001 | 4 (−1, 9) | 7.5 | 0.11 |
| 5-min HRR | −55 (−52, −57) | −60 (−58, −63) | 6 (3, 8) | 10.9 | <0.0001 | 4 (−1, 9) | 7.1 | 0.14 |
| **Aerobic + Resistance** | | | | | | | | |
| 1-min HRR | −26 (−23, −30) | −29 (−25, −33) | 2 (1, 6) | 7.7 | 0.20 | 4 (−2, 10) | 16.0 | 0.19 |
| 2-min HRR | −42 (−38, −46) | −47 (−43, −51) | 5 (1, 9) | 11.9 | 0.01 | 4 (−2, 10) | 9.3 | 0.22 |
| 3-min HRR | −50 (−46, −54) | −56 (−52, −60) | 6 (2, 10) | 12.0 | 0.003 | 6 (−0.4, 12) | 12.0 | 0.07 |
| 4-min HRR | −53 (−49, −57) | −59 (−55, −63) | 6 (2, 10) | 11.3 | 0.002 | 6 (−0.1, 12) | 11.3 | 0.06 |
| 5-min HRR | −56 (−52, −60) | −62 (−58, −66) | 6 (2, 10) | 10.7 | 0.002 | 6 (−1, 12) | 10.7 | 0.08 |

Data are presented as adjusted least square means (95% CI).

Model was adjusted for sex, Tanner stage, ethnicity, training duration, and types of exercise.

HRR, heart rate recovery.

## Associations between changes in HRR, body composition and metabolic risk factors after exercise intervention

There was no association between the change in HRR with the change in waist circumference or % body fat with or without adjustment for covariates and intervention group (Table 3, *P* > 0.05). A positive relationship between the change in 1-min HRR and the change in body weight was observed after adjusting for covariates and the exercise intervention group (*P* = 0.01). No association was observed between the change in HRR with the change in total cholesterol, triglycerides, HDL cholesterol, LDL cholesterol, or VLDL cholesterol with or without adjustment for covariates and intervention group (Table 3, *P* > 0.05).

## Discussion

The principal findings are that AE, RE, and combined AE and RE training are associated with significant improvements in HRR after 3–6 months of training. However, only the improvements in HRR in the AE group were significantly greater

**Table 3. Associations between changes in heart rate recovery, VO$_{2peak}$, body composition and cardiometabolic risk factors in adolescents with overweight or obesity after 3 or 6 months of aerobic exercise, resistance exercise, or combined aerobic and resistance exercise.**

| | 1-min HRR | | | 2-min HRR | | | 3-min HRR | | | 4-min HRR | | | 5-min HRR | | |
|---|---|---|---|---|---|---|---|---|---|---|---|---|---|---|---|
| | β | Std Error | P value | β | Std Error | P value | β | Std Error | P value | β | Std Error | P value | β | Std Error | P value |
| VO$_2$peak (mL/kg/min) | −0.31 | 0.25 | 0.22 | −1.06 | 0.21 | <0.001 | −1.24 | 0.18 | <0.001 | −1.21 | 0.16 | <0.001 | −1.17 | 0.15 | <0.001 |
| Body Weight (kg) | 0.63 | 0.25 | 0.01 | 0.43 | 0.23 | 0.07 | 0.34 | 0.21 | 0.11 | 0.31 | 0.20 | 0.12 | 0.33 | 0.19 | 0.08 |
| Waist Circumference (cm) | 0.48 | 0.32 | 0.14 | 0.15 | 0.30 | 0.61 | 0.02 | 0.27 | 0.94 | −0.08 | 0.25 | 0.76 | −0.09 | 0.24 | 0.70 |
| Body Fat (%) | 0.78 | 0.57 | 0.17 | 0.68 | 0.52 | 0.20 | 0.70 | 0.47 | 0.14 | 0.83 | 0.43 | 0.06 | 0.63 | 0.42 | 0.13 |
| Total Cholesterol (mg/dL) | 0.01 | 0.07 | 0.93 | 0.003 | 0.06 | 0.96 | 0.01 | 0.05 | 0.91 | −0.01 | 0.05 | 0.81 | −0.004 | 0.05 | 0.94 |
| Triglycerides (mg/dL) | 0.0003 | 0.01 | 0.98 | −0.01 | 0.01 | 0.42 | −0.003 | 0.01 | 0.84 | −0.01 | 0.01 | 0.39 | −0.003 | 0.01 | 0.79 |
| HDL Cholesterol (mg/dL) | −0.39 | 0.21 | 0.07 | −0.06 | 0.20 | 0.75 | −0.10 | 0.18 | 0.58 | −0.19 | 0.17 | 0.25 | −0.27 | 0.16 | 0.09 |
| LDL Cholesterol (mg/dL) | 0.06 | 0.09 | 0.51 | 0.05 | 0.08 | 0.56 | 0.03 | 0.07 | 0.72 | 0.03 | 0.07 | 0.60 | 0.02 | 0.06 | 0.75 |
| VLDL Cholesterol (mg/dL) | 0.002 | 0.07 | 0.98 | −0.05 | 0.07 | 0.42 | −0.01 | 0.06 | 0.84 | −0.05 | 0.06 | 0.39 | −0.01 | 0.05 | 0.79 |

All models were adjusted for sex, Tanner stage, ethnicity, training duration, and types of exercise.

VO$_2$peak, peak oxygen uptake; HDL, high-density lipoprotein; LDL, low-density lipoprotein; VLDL, very low-density lipoprotein.

when compared to the control group. These findings suggest that AE training is an effective strategy for improving HRR in previously sedentary adolescents with overweight or obesity.

To our knowledge, this is the first study to examine the effects of AE, RE and combined AE and RE training on HRR in adolescents with overweight or obesity. Previous studies have observed improvements in HRR following AE training in adults with overweight or obesity [16–19]. Kim et al. [17] demonstrated that 12-weeks of moderate-intensity AE training (60–70% of maximal HR, 60 minutes/day, 3 days/week) without calorie restriction significantly improved HRR following a symptom-limited bicycle exercise test in middle-aged men with obesity and metabolic syndrome, but not those without metabolic syndrome. However, this study did not have a control group. Similarly, in middle-aged women with obesity, Chaudhary et al. [19] have shown significant improvements in HRR following 6-weeks of moderate-intensity AE (3 days/week at 60–70% of maximum HR) without dietary restriction, but not resistance training (7 different types of exercises, 4 sets of 10 repetitions/set of each exercise), but did not assess if the changes were different relative to control. In adolescents with overweight and obesity, we observed reductions in HRR within all exercise groups. However, only the AE group showed a significantly faster HRR compared to the control group. It is unclear whether the lack of differences in the combined AE and RE group is due to the lack of beneficial effects of RE for HRR or if the lower AE dose in the combined group failed to reach the threshold necessary for improvements in HRR. Together, these findings suggest that AE training may be more beneficial than RE training or combined AE and RE training for improving HRR in adolescents with overweight or obesity.

Reduced parasympathetic activation during the first minute following exercise cessation is considered to be responsible for impaired 1-min HRR [28,29]. Meanwhile, a delayed HRR during 2- to 5-min post-exercise may be caused by reduced parasympathetic activation and a slow withdrawal of sympathetic activity through the delayed clearance of catecholamines, such as norepinephrine [28,29]. AE training can enhance the balance and response of parasympathetic reactivation and sympathetic activity, resulting in a faster HRR after exercise [28,29]. Increased vagal modulation has been shown following AE, while reduced vagal modulation has been observed following RE [30–32], which may explain why greater improvements in HRR were observed following AE training in our study. There is some evidence to suggest that the upregulation of nitric oxide synthase 1 (NOS-1) in intracardiac parasympathetic ganglia may play a key role in mediating the improvements in parasympathetic function following AE training [33,34].

Previous research has demonstrated that a slow HRR (≤12 bpm at 1-min HRR) following maximal exercise is associated with an elevated risk of mortality in adults [5]. Children have been shown to display a faster 1-min HRR for a given exercise intensity than adults [35], and in the current study, the mean 1-min HRR at baseline in all exercise intervention

groups was greater than 20 bpm. Although these values are higher than the adult threshold, HRR has been shown to decline with age [36]. By improving HRR earlier in life, it may delay the decrease in HRR that can occur in individuals with overweight and obesity.

In the current study, there was a negative relationship between the improvements in 2-, 3-, 4-, and 5-min HRR with the improvements in $VO_{2peak}$ after accounting for covariates and the exercise training group. This suggests that the improvements in $VO_{2peak}$ that occurs with exercise training is more closely related to the parasympathetic reactivation and sympathetic withdrawal that is associated with HRR at 2–5 min post-exercise rather than the parasympathetic reactivation alone that occurs during the first minute of HRR. Further, these findings strengthen the use of $VO_{2peak}$ as an indicator for predicting cardiovascular disease risk in adolescents with overweight or obesity. Developing exercise training programs that emphasize improvements in $VO_{2peak}$ may lead to greater improvements in HRR and a reduction in cardiovascular disease risk in this population.

Our findings further demonstrate that there was no association between the change in HRR and the change in body weight, % body fat or cardiometabolic risk factors. Similarly, a previous study involving combined AE and RE did not observe significant associations between the change in 1-min HRR with the change in obesity indices or cardiometabolic risk factors in children and adolescents [37]. Findings from cross-sectional research have displayed an association of HRR with various obesity traits and cardiometabolic factors in children (8–10 years), but no associations were observed in adolescents (14–16 years) [38]. This suggests that the physiological and hormonal changes occurring during adolescence may be concealing the association of HRR with these risk factors [38]. Further research is required to confirm these hypotheses.

The strengths and limitations of this study warrant mention. A strength of this study is that we compared HRR between different exercise interventions in a large sample of adolescents with overweight or obesity. We also employed the criterion methods of evaluating cardiorespiratory fitness. However, the study sample consisted of adolescents primarily of black or white ethnicity. Consequently, the findings of this study may not be generalizable to adolescents in other ethnic groups. While HRR has been validated as a measure of cardiac autonomic function [39], it would be useful to confirm these findings using other measures of autonomic function, such as heart rate variability.

In conclusion, our findings suggest that AE training is an effective strategy for improving HRR in adolescents with overweight or obesity. Improvements in HRR are associated with the improvements in $VO_{2peak}$ following exercise training, but does not appear to be associated with changes in obesity or cardiometabolic risk factors.

## Supporting information

**S1 File. PLOSOne_Human_Subjects_Research_Checklist (1).**
(DOCX)

## Acknowledgments

The authors would like to express their gratitude to the study participants.

## Author contributions

**Conceptualization:** Rubin Pooni, Silva Arslanian, SoJung Lee, Jennifer L. Kuk.

**Data curation:** Rubin Pooni, SoJung Lee.

**Formal analysis:** Rubin Pooni.

**Funding acquisition:** Silva Arslanian, SoJung Lee.

**Investigation:** Silva Arslanian, SoJung Lee, Jennifer L. Kuk.

**Methodology:** Heather Edgell, Hala Tamim, SoJung Lee, Jennifer L. Kuk.

**Project administration:** Silva Arslanian, SoJung Lee, Jennifer L. Kuk.

**Resources:** Silva Arslanian, SoJung Lee, Jennifer L. Kuk.

**Supervision:** Silva Arslanian, Heather Edgell, Hala Tamim, SoJung Lee, Jennifer L. Kuk.

**Writing – original draft:** Rubin Pooni.

**Writing – review & editing:** Silva Arslanian, Heather Edgell, Hala Tamim, SoJung Lee, Jennifer L. Kuk.

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
