## [Decision Letter · Decision Letter 0]

1 Jul 2025

Dear Dr. Kuk,

Thank you for submitting your manuscript to PLOS ONE. After careful consideration, we feel that it has merit but does not fully meet PLOS ONE’s publication criteria as it currently stands. Therefore, we invite you to submit a revised version of the manuscript that addresses the points raised during the review process.

**ACADEMIC EDITOR: ** Dear Author, after peer review, we believe your study addresses an important topic, but it requires major revisions before it can be considered for publication.

Please carefully address all reviewer comments and provide a point-by-point response with your revised manuscript. 

We look forward to receiving your revised manuscript.

Kind regards,

Zulkarnain Jaafar

Academic Editor

PLOS ONE

Journal Requirements: 

 [The current analyses received no specific funding. The original research was funded by the National Institutes of Health (5R01HL114857, 1R21DK083654-01A1), American Diabetes Association (7-08-JF-27), UPMC Children’s Hospital of Pittsburgh (Cochrane-Weber Foundation and Renziehausen Fund) to Lee, and the National Center for Advancing Translational Sciences Clinical and Translational Science Award (UL1 RR024153, UL1TR000005) to the Pediatric Clinical and Translational Research Center at UPMC Children’s Hospital of Pittsburgh.]. 

3. In the online submission form, you indicated that [Data will be shared by Dr. Lee with ethical approval from their relevant ethical review board.].

Reviewers' comments:

Reviewer's Responses to Questions

**Comments to the Author**

1. Is the manuscript technically sound, and do the data support the conclusions?

Reviewer #1: Partly

2. Has the statistical analysis been performed appropriately and rigorously?

Reviewer #1: Yes

3. Have the authors made all data underlying the findings in their manuscript fully available?

Reviewer #1: No

4. Is the manuscript presented in an intelligible fashion and written in standard English?

Reviewer #1: Yes

Reviewer #1: The manuscript titled “Effects of different types of exercise training on heart rate recovery in adolescents with overweight and obesity” addresses a timely and clinically relevant topic. It investigates the comparative effects of aerobic (AE), resistance (RE), and combined training on heart rate recovery (HRR), while also examining the association between HRR and both body composition and cardiometabolic risk factors. This dual focus greatly enhances the study’s clinical and translational relevance, particularly given the rising prevalence of obesity-related health issues in adolescents and the increasing recognition of HRR as an early marker of autonomic and cardiovascular function. Additionally, the inclusion of cardiometabolic and body composition variables allows for a more comprehensive interpretation of physiological adaptations to exercise.

To strengthen the clarity and impact of the manuscript, several points require revision or elaboration. First, while the current title reflects the central focus on exercise and HRR, it does not fully capture the study's dual aims. I recommend revising the title to: “Effects of Exercise Modalities on Heart Rate Recovery and Its Association with Cardiometabolic Risk in Overweight and Obese Adolescents.” This version more clearly communicates both the interventional and correlational aspects of the study and will likely increase the manuscript’s appeal to a broader readership.

Second, to enhance the transparency and interpretability of your statistical findings, it is recommended that percentage changes from pre- to post-intervention be reported alongside exact p-values and effect sizes (e.g., partial eta squared, η²). This is particularly important given the use of repeated measures ANCOVA, where understanding the magnitude and clinical relevance of observed changes is just as important as their statistical significance. Consistently including these values in the abstract, results section, and tables would significantly improve the scientific rigor of the paper.

A key concern lies in the variation of intervention duration across groups: while the control group and some exercise groups were monitored for three months, the combined training group received a six-month intervention. Such inconsistency introduces a potential confounder that can affect the comparability of results across exercise modalities. Although statistical adjustment for time may mitigate this to some extent, a clearer justification for the differing durations, or better yet, standardization across groups, would strengthen the internal validity of the study. The authors are encouraged to either equalize the intervention periods or provide robust reasoning and analytical adjustment for this variability.

Further improvements can be made in the methodology section by specifying the brand name, manufacturer, and country of origin for all key measurement equipment, including those used for anthropometric assessments, cardiometabolic profiling, and exercise testing. Moreover, referencing validation and reliability data from previous studies for each device would enhance the study's methodological transparency and replicability. This detail is important for future researchers aiming to reproduce or build upon your work.

Regarding the presentation of findings, the results section would benefit from more structured and reader-friendly data presentation. Currently, pre–post and between-group effects are described narratively, but without accompanying summary tables. To enhance clarity, I strongly recommend including two tables: one summarizing within-group changes, and another presenting between-group differences with relevant statistical indicators such as mean change, percent change, confidence intervals, p-values, and effect sizes. These tables would provide a more accessible overview of key outcomes and help readers better understand the intervention effects.

In the discussion section, the physiological rationale behind the observed changes in HRR should be expanded. While parasympathetic reactivation and sympathetic withdrawal are briefly mentioned, the authors should elaborate on how these mechanisms are differentially influenced by aerobic versus resistance training. For instance, aerobic training is known to induce specific adaptations in autonomic tone and vagal modulation, which should be explicitly connected to the improvements observed in HRR. Additionally, the reported associations between HRR and VO₂peak represent an important finding, but the manuscript does not fully explore their implications. It would strengthen the discussion to elaborate on how these relationships could inform future exercise prescriptions, risk stratification, or preventive strategies for cardiovascular disease in adolescent populations.

Overall, this is a valuable and well-conceived study with potential for significant clinical and scientific impact. Addressing the methodological and interpretive issues raised above will enhance the quality, credibility, and applicability of the manuscript.

**Do you want your identity to be public for this peer review?** For information about this choice, including consent withdrawal, please see our Privacy Policy

Reviewer #1: **Yes: ** Friew Amare

---

## [Author Response · Author response to Decision Letter 1]

16 Sep 2025

Reviewer #1

Comment 1: To strengthen the clarity and impact of the manuscript, several points require revision or elaboration. First, while the current title reflects the central focus on exercise and HRR, it does not fully capture the study's dual aims. I recommend revising the title to: “Effects of Exercise Modalities on Heart Rate Recovery and Its Association with Cardiometabolic Risk in Overweight and Obese Adolescents.” This version more clearly communicates both the interventional and correlational aspects of the study and will likely increase the manuscript’s appeal to a broader readership.

Response: Thank you, we have revised the title as suggested by the reviewer, but have slightly modified the wording to use person first language for obesity. The revised title now reads: Effects of Exercise Modalities on Heart Rate Recovery and Its Association with Cardiometabolic Risk in Adolescents with Overweight and Obesity

Comment 2: Second, to enhance the transparency and interpretability of your statistical findings, it is recommended that percentage changes from pre- to post-intervention be reported alongside exact p-values and effect sizes (e.g., partial eta squared, η²). This is particularly important given the use of repeated measures ANCOVA, where understanding the magnitude and clinical relevance of observed changes is just as important as their statistical significance. Consistently including these values in the abstract, results section, and tables would significantly improve the scientific rigor of the paper.

Response: As suggested, we have added least squared adjusted mean changes with 95% CI, percentage changes, and p-values for pre- to post-intervention changes in HRR have been added to the abstract, results section, and a new table (Table 2). Unfortunately, we were unable to add partial eta squared values for the within person repeated measures analyses as proc mixed in SAS does not have the effect size option and does not give us the necessary values to manually compute it. Proc mixed is preferred for repeated measures as it better handles the correlated observations and accounts for within subject covariability, and has better flexibility in the model to compare within subject effect contrasts. We hope that the inclusion of the least squared adjusted mean changes with 95% CI will help the reader understand the effect size which is essentially the size of the change relative to the variability of the change.

Comment 3: A key concern lies in the variation of intervention duration across groups: while the control group and some exercise groups were monitored for three months, the combined training group received a six-month intervention. Such inconsistency introduces a potential confounder that can affect the comparability of results across exercise modalities. Although statistical adjustment for time may mitigate this to some extent, a clearer justification for the differing durations, or better yet, standardization across groups, would strengthen the internal validity of the study. The authors are encouraged to either equalize the intervention periods or provide robust reasoning and analytical adjustment for this variability.

Response: We understand the variation in intervention durations is a potential confounder. However, no main effect or interaction effect for intervention duration was observed in our models, which is why the interventions were combined for analysis. We have added this information to the Statistical Analyses section within the Methods (Page 8).

Comment 4: Further improvements can be made in the methodology section by specifying the brand name, manufacturer, and country of origin for all key measurement equipment, including those used for anthropometric assessments, cardiometabolic profiling, and exercise testing. Moreover, referencing validation and reliability data from previous studies for each device would enhance the study's methodological transparency and replicability. This detail is important for future researchers aiming to reproduce or build upon your work.

Response: Thank you. We have added the make and model for the anthropometric (page 6) and VO2peak test equipment (page 7), but do not have that information for the cardiometabolic data. All analyses for fasting plasma lipids were measured in the Nutrition Laboratory of the University of Pittsburgh Graduate School of Public Health, which was certified by the Centers for Disease Control and Prevention-National Heart, Lung, and Blood Institute Lipid Standardization Program. We have also included coefficients of variations for cardiometabolic risk factors (Page 6) in the revised manuscript.

Comment 5: Regarding the presentation of findings, the results section would benefit from more structured and reader-friendly data presentation. Currently, pre–post and between-group effects are described narratively, but without accompanying summary tables. To enhance clarity, I strongly recommend including two tables: one summarizing within-group changes, and another presenting between-group differences with relevant statistical indicators such as mean change, percent change, confidence intervals, p-values, and effect sizes. These tables would provide a more accessible overview of key outcomes and help readers better understand the intervention effects.

Response: Thank you. We have revised the results as suggested as in combination with the suggestions in comment 3. Please see our response to comment 3.

Comment 6: In the discussion section, the physiological rationale behind the observed changes in HRR should be expanded. While parasympathetic reactivation and sympathetic withdrawal are briefly mentioned, the authors should elaborate on how these mechanisms are differentially influenced by aerobic versus resistance training. For instance, aerobic training is known to induce specific adaptations in autonomic tone and vagal modulation, which should be explicitly connected to the improvements observed in HRR. Additionally, the reported associations between HRR and VO2peak represent an important finding, but the manuscript does not fully explore their implications. It would strengthen the discussion to elaborate on how these relationships could inform future exercise prescriptions, risk stratification, or preventive strategies for cardiovascular disease in adolescent populations.

Response: The difference in vagal modulation following aerobic versus resistance exercise training, the strengthened use of VO2peak as an indicator of CVD risk, and a recommended emphasis on VO2peak improvements in future exercise training programs have been added to the Discussion (Page 13 and Page 14). 

Reviewer #2

Comment 1: The current title provides a fundamental overview of the study's emphasis but does not fully reflect the dual aims mentioned in the manuscript, namely the assessment of the association between heart rate recovery (HRR) and both body composition and cardiometabolic risk factors. To clarify the study's scope, I recommend changing the title to "Effects of Exercise Modalities on Heart Rate Recovery and Its Association with Cardiometabolic Risk in Overweight and Obese Adolescents."

The revised title clearly states the intervention's comparative character as well as the correlational analysis including HRR and cardiometabolic outcomes, boosting both specificity and relevance for potential readers.

Response: Thank you, we have revised the title as suggested by the reviewer, but have slightly modified the wording to use person first language for obesity. The revised title now reads: Effects of Exercise Modalities on Heart Rate Recovery and Its Association with Cardiometabolic Risk in Adolescents with Overweight and Obesity

Comment 2: To strengthen the clarity and interpretability of your findings, I recommend that you include percentage change for pre- to post-test analyses, in addition to reporting exact p-values and effect sizes (e.g., partial eta squared, η²) for all outcomes where significant differences were found. Since your analysis is based on repeated measures ANCOVA, these additional statistical details are crucial for conveying not only statistical significance but also the magnitude and practical relevance of the observed effects. Consistent reporting of these values in both the abstract and the main results section will enhance the scientific rigor, transparency, and overall impact of your manuscript.

Response: We have added least squared adjusted mean changes with 95% CI, percentage changes, and p-values for pre- to post-intervention changes in HRR have been added to the abstract, results section, and a new table (Table 2). Unfortunately, we were unable to add partial eta squared values for the within person repeated measures analyses as proc mixed in SAS does not have the effect size option and does not give us the necessary values to manually compute it. Proc mixed is preferred for repeated measures as it better handles the correlated observations and accounts for within subject covariability, and has better flexibility in the model to compare within subject effect contrasts. We hope that the inclusion of the least squared adjusted mean changes with 95% CI will help the reader understand the effect size which is essentially the size of the change relative to the variability of the change.

Comment 3: In the methodology section, it is noted that the control group was observed for three months, while participants in the AE and RE groups underwent intervention for either three or six months, and the combined training (CT) group participated for the full six months. This variation in intervention duration across groups raises concerns regarding the comparability of exercise modalities. Even if time is statistically treated as a confounding variable, the unequal exposure duration introduces a potential bias that may affect the interpretation of outcomes. To ensure valid comparisons between exercise types, intervention duration should ideally be standardized across groups or more clearly justified and accounted for in the analysis and discussion.

Response: We understand the variation in intervention durations is a potential confounder. However, no main effect or interaction effect for intervention duration was observed in our models, which is why the interventions were combined for analysis. We have added this information to the Statistical Analyses section within the Methods (Page 8).

Comment 4: To further demonstrate the rigorous scientific methodology and replicability of your study, it is recommended that you provide the brand name, manufacturer, and country of origin for all essential equipment utilized, such as those for height, weight, lipid profile measurement, and treadmill-based testing or training. Please add information about the validity and reliability of these instruments from past validation studies. This information is critical for assessing measurement accuracy and consistency, as well as allowing other researchers to reproduce your processes or compare results across studies. Incorporating these elements within the Methods section will increase the scientific credibility of your work.

Response: Thank you. We have added the make and model for the anthropometric equipment, but do not have that information for the cardiometabolic data. All analyses for fasting plasma lipids were measured in the Nutrition Laboratory of the University of Pittsburgh Graduate School of Public Health, which was certified by the Centers for Disease Control and Prevention-National Heart, Lung, and Blood Institute Lipid Standardization Program. This should mean that the results obtained should be the same regardless of the equipment used.

Comment 5: As the primary objective of your study is to examine the effects of different types of exercise training on heart rate recovery, it is important to clearly present both within-group (pre–post) and between-group effects in the Results section. While the manuscript discusses pre–post analyses result and between group effect in the text, there is currently no table provided that summarizes these findings. To improve the clarity, transparency, and interpretability of your results, I strongly recommend including two tables: one showing pre–post changes within each group, and another summarizing the between-group effects, with appropriate statistical metrics such as percent change, mean differences, confidence intervals, p-values, and effect sizes (e.g., η²). This will significantly enhance the readability and scientific rigor of your manuscript.

Response: Please see our response to comment 3

Comment 6: In the Discussion section, the physiological rationale for the observed improvement in heart rate recovery (HRR) warrants further elaboration. While parasympathetic reactivation and sympathetic withdrawal are briefly mentioned, the authors should more thoroughly explain these mechanisms and explicitly relate them to the specific exercise interventions used. For instance, detailing how aerobic exercise uniquely promotes autonomic adaptations would provide a clearer interpretation of the findings. Integrating this explanation would enhance the discussion by linking the results to underlying physiological processes and aligning them with existing research. And also the reported association between improvements in HRR (at 2–5 minutes) and VO2peak is an important finding, yet its practical significance is not fully elaborated. How might these relationships inform future training prescriptions or screening strategies for cardiovascular risk in adolescents? Including such implications would enhance the translational value of the findings.

Response: The difference in vagal modulation following aerobic versus resistance exercise training, the strengthened use of VO2peak as an indicator of CVD risk, and a recommended emphasis on VO2peak improvements in future exercise training programs have been added to the Discussion.

---

## [Decision Letter · Decision Letter 1]

19 Oct 2025

Dear Dr. Kuk,

Thank you for submitting your manuscript to PLOS ONE. After careful consideration, we feel that it has merit but does not fully meet PLOS ONE’s publication criteria as it currently stands. Therefore, we invite you to submit a revised version of the manuscript that addresses the points raised during the review process.

**ACADEMIC EDITOR: **

We look forward to receiving your revised manuscript.

Kind regards,

Zulkarnain Jaafar

Academic Editor

PLOS ONE

Journal Requirements:

Reviewers' comments:

Reviewer's Responses to Questions

**Comments to the Author**

Reviewer #1: All comments have been addressed

Reviewer #2: (No Response)

2. Is the manuscript technically sound, and do the data support the conclusions?

Reviewer #1: Yes

Reviewer #2: Yes

3. Has the statistical analysis been performed appropriately and rigorously?

Reviewer #1: Yes

Reviewer #2: Yes

4. Have the authors made all data underlying the findings in their manuscript fully available?

Reviewer #1: Yes

Reviewer #2: No

5. Is the manuscript presented in an intelligible fashion and written in standard English?

Reviewer #1: Yes

Reviewer #2: Yes

Reviewer #1: Thank you for the clarification. However, I still have some reservations regarding the instruments used, particularly for the cardiometabolic data. While certification of the laboratory by the CDC–NHLBI Lipid Standardization Program ensures quality control, it would still be helpful to provide at least general information about the analytical instruments or assay methods used (e.g., type of analyzer, assay principle, or manufacturer). This additional detail would further enhance methodological transparency and allow for better reproducibility and comparison across studies.

Reviewer #2: Many thanks for the invitation to review this manuscript. In this paper the authors evaluate the effect of exercise modalities on heart rate recovery and its association with cardiometabolic risk in adolescents with overweight or obesity. The paper is well written and presents good elements of originality. I have no further comments.

**Do you want your identity to be public for this peer review?** For information about this choice, including consent withdrawal, please see our Privacy Policy

Reviewer #1: **Yes: ** Friew Amare Mengistu

Reviewer #2: No

---

## [Author Response · Author response to Decision Letter 2]

21 Oct 2025

Reviewer #1:

Thank you for the clarification. However, I still have some reservations regarding the instruments used, particularly for the cardiometabolic data. While certification of the laboratory by the CDC–NHLBI Lipid Standardization Program ensures quality control, it would still be helpful to provide at least general information about the analytical instruments or assay methods used (e.g., type of analyzer, assay principle, or manufacturer). This additional detail would further enhance methodological transparency and allow for better reproducibility and comparison across studies.

Response: We have elaborated on the methods used to assess the lipids in the manuscript methods as follows:

“A fasting blood sample was obtained for lipid analysis following a minimum 10-hour overnight fast. Fasting plasma lipids were measured using an Olympus AU400 Chemistry Analyzer in the Nutrition Laboratory of the University of Pittsburgh Graduate School of Public Health, which was certified by the Centers for Disease Control and Prevention-National Heart, Lung, and Blood Institute Lipid Standardization Program [26]. HDL was assessed using the two-reagent method. Triglycerides were determined enzymatically. Reagents used were obtained from Vital Diagnostics (Lincoln, RI). LDL and VLDL were calculated using the Friedewald equation [27]. For total and HDL cholesterol and triglycerides intra-assay coefficients of variation (CV) were 1.0%, 1.8% and 1.8% and inter-assay CV were 1.6%, 2.6%, and 3.7%, respectively.”

Reviewer #2:

Many thanks for the invitation to review this manuscript. In this paper the authors evaluate the effect of exercise modalities on heart rate recovery and its association with cardiometabolic risk in adolescents with overweight or obesity. The paper is well written and presents good elements of originality. I have no further comments.

Response: Thank you. We appreciate your contributions in this process to improve our manuscript.

---

## [Decision Letter · Decision Letter 2]

26 Oct 2025

Effects of Exercise Modalities on Heart Rate Recovery and Its Association with Cardiometabolic Risk in Adolescents with Overweight or Obesity

PONE-D-25-26093R2

Dear Dr. Kuk,

We’re pleased to inform you that your manuscript has been judged scientifically suitable for publication and will be formally accepted for publication once it meets all outstanding technical requirements.

Kind regards,

Zulkarnain Jaafar

Academic Editor

PLOS ONE

Additional Editor Comments (optional):

Reviewers' comments:

Reviewer's Responses to Questions

**Comments to the Author**

Reviewer #1: All comments have been addressed

2. Is the manuscript technically sound, and do the data support the conclusions?

Reviewer #1: Yes

3. Has the statistical analysis been performed appropriately and rigorously?

Reviewer #1: Yes

4. Have the authors made all data underlying the findings in their manuscript fully available?

Reviewer #1: Yes

5. Is the manuscript presented in an intelligible fashion and written in standard English?

Reviewer #1: Yes

Reviewer #1: (No Response)

**Do you want your identity to be public for this peer review?** For information about this choice, including consent withdrawal, please see our Privacy Policy

Reviewer #1: **Yes: ** Friew Amare Mengistu

---

## [Editor Report · Acceptance letter]

PONE-D-25-26093R2

PLOS ONE

Dear Dr. Kuk,

I'm pleased to inform you that your manuscript has been deemed suitable for publication in PLOS ONE. Congratulations! Your manuscript is now being handed over to our production team.

Kind regards,

on behalf of

Dr. Zulkarnain Jaafar

Academic Editor

PLOS ONE